# An Updated Reconstruction of Antarctic Near-Surface Air Temperatures at Monthly Intervals Since 1958

David Bromwich[1], Sheng-Hung Wang[1], Xun Zou[2], Alexandra Ensign[1].

[1]Polar Meteorology Group, Byrd Polar and Climate Research Center, The Ohio State University, Columbus, Ohio 43210, USA.
[2]Center for Western Water and Weather Extremes, Scripps Institution of Oceanography, La Jolla, California 92037, USA.

*Correspondence to*: David Bromwich (bromwich.1@osu.edu)

**Abstract.** An updated near-surface temperature reconstruction for the Antarctic continent is presented for 1958 to 2022 (65 years) as monthly anomalies relative to 1981-2010 (RECON; Bromwich and Wang, 2024). It is based on monthly mean 2-m temperatures at 15 fixed stations that are spatially extrapolated to the entire continent using weights derived from the European Centre for Medium-Range Weather Forecasts 5th generation reanalysis (ERA5) and has a grid spacing of 60 km. Infilling of the fixed station records are performed where necessary to yield complete time series for 1958-2022. Variability and trends are tested at independent stations that have much shorter periods of record. RECON is designed for Antarctic climate variability and change applications for large spatial scales and extended time scales.

## 1. Introduction

Near-surface air temperature is a fundamental variable for describing and understanding the climate. For those regions of Earth that are remote and sparsely populated, establishing their temperature history from direct observations can be a major challenge. For Antarctica, a substantial network of permanent research stations was established in conjunction with the International Geophysical Year (IGY) of 1957-1958 (e.g., Jones et al., 2019), although there are isolated sites in the Antarctic Peninsula region that predate the IGY. One of these is Orcadas station that has the longest continuous temperature record south of 60°S and which started in 1903. As a result, the derivation of the continental temperature regime from meteorological observations should start from the IGY. One complication is that the temperature records were collected for a variety of applications, most often for weather forecasting purposes, and the quality is not always suitable for detecting small climate changes. So, care is needed in applying these data to derive Antarctica's air temperature history, e.g., Lazzara et al. (2012).

Previous reconstructions of Antarctic temperatures were performed in the context of coarse-resolution global analyses such as from GIS-Temp (Lenssen et al., 2024), HadCRUT5 (Morice et al., 2021), NOAA GlobalTemp (Yin et al., 2024) and Berkeley Earth (Rohde and Hausfather, 2020). All available observations were utilized, and results were extrapolated into data sparse regions typically using kriging-type analysis. Error estimates were provided. In contrast, this Antarctic focused temperature reconstruction employs vetted observations from 15 fixed stations along

with a kriging extrapolation to fill in data void regions. Independent observations are used to establish the reliability
of this (much) higher resolution reconstruction.

The manuscript is organized as follows. Section 2 summarizes the temperature observations applied and the spatial
extrapolation approach. This is followed by performance testing of the resulting temperature reconstruction against
independent observations that extend for at least 10 years and are selected to sample the range of Antarctic
environments. Section 4 presents an example application of the reconstruction. Sections 5 and 6 respectively provide
data availability and a discussion.

**2. Methodology**

One key consideration for reconstructing the continental temperature from station observations is the spatial
extrapolation from these point observations to the entire continent. For this task, we depend on global reanalyses that
reconstruct the weather and climate across the entire Earth from a wide variety of meteorologically related
observations. For Antarctica and the Southern Ocean, such reanalyses have much lower quality prior to 1979 when
there was very limited satellite coverage over the data sparse Southern Ocean (e.g., Bromwich et al., 2024). So, our
use of global reanalyses for spatial temperature extrapolation is restricted to after 1979, and even then spurious
features such as artificial trends can be present. We employ temperature anomalies that the reanalyes tend to
skillfully capture and that typically have a large spatial footprint especially for interior Antarctica (e.g., Zhu et al.,
2021, Fig. 3; King et al., 2003). It is assumed that the spatial relationships for the temperature anomalies established
from ERA5 apply to the entire observational record.

Nicolas and Bromwich (2014) reconstructed the air temperature over Antarctica from monthly temperature
observations at 15 fixed stations across Antarctica on a 60-km polar stereographic projection. These data were
spatially extrapolated to the entire continent based on the statistical linkages between the stations and all grid points
covering Antarctica from the Climate Forecast System Reanalysis (CFSR) for the 30-year period 1979-2009. The
reconstruction closely matched the station observations, was not impacted by anomalous temperature trends in the
reanalysis and was verified against independent temperature observations. It spanned 1958-2012 at monthly
intervals. We present a new version of this data set in this manuscript.

To revise and update the Nicolas and Bromwich (2014) analysis, Belgrano Station is employed instead of Halley
Station. This is done because the frequent relocation of the Halley observation site on the floating Brunt Ice Shelf
led to artifacts in the temperature time series resulting in weak cooling for 1957-2019 whereas weak warming likely
occurred (King et al., 2021). The other 14 stations used by Nicolas and Bromwich (2014) are applied here. Figure 1
locates these sites along with Halley that is replaced by Belgrano. The European Centre for Medium-Range Weather
Forecasts (ECMWF) 5[th] Generation Reanalysis (ERA5; Hersbach et al., 2020) is employed to provide the spatial
weights that extrapolate the station observations. ERA5 is a more modern and higher resolution global reanalysis
than CFSR and has fewer issues with anomalous temperature trends (Gossart et al., 2019), although it has a several
degree warm bias during winter in the Antarctic interior. Testing for the 1958-2012 period using the 15 stations
employed by Nicolas and Bromwich (2014) demonstrated that CFSR and ERA5 based spatial extrapolation
produced very similar results (not shown). Further, Nicolas and Bromwich (2014) and Screen and Simmonds (2012)
found that spatial extrapolation to the entire continent from long-term station observations was relatively insensitive
to the reanalysis used for near surface air temperatures and free atmosphere temperatures, respectively.

Monthly average 2-m temperatures from the 14 stations employed by Nicolas and Bromwich (2014) as well as
Belgrano (Fig. 1) were updated through 2022. Table 1 describes the sources used for the updates (primarily the
READER site; Turner et al., 2004; but supplemented by several more reliable records), and the steps employed fill
in the gaps that were present in the data from READER. The reconstructed Byrd Station record is based on
Bromwich et al. (2013, 2014). The Belgrano Station record is another reconstruction that requires detailed
discussion. The 1958-1960 values at Belgrano II (actual observations 1980-present) were based on Belgrano I 1958-
1960 READER observations estimated for Belgrano II location by employing Halley Station monthly temperature
observations that were available for both 1958-1960 as well as when Belgrano II was in operation. For 1961-1979 at
Belgrano II, we used estimates produced by the Global Historical Climatology Network – Monthly Mean
Temperature Version 4 (Menne et al., 2018), denoted as GHCNm version 4 QFE. Menne et al. (2018, p. 9847)
outlined that the estimation procedure "iterates to find a set of neighboring correlated series for each station series
requiring estimates (the target) that minimizes the confidence limits for the difference between the target values and
estimates of these values derived using neighboring values. The difference between the target and neighbor average
is used as an offset in the interpolation to account for climatological differences between the target and neighbors."
For 1980 and later, the Belgrano II record provided by READER has 75% or more of the 6 hourly observations for
each month that we adopt as a sound basis for computing reliable monthly mean temperatures.  GHCNm version 4
QFE values are used for 4 missing data periods in 1980, 1981, 2002, and 2003 at Belgrano II. Other notable aspects
from Table 1 are that GCHNm quality-controlled and adjusted values (GHCNm version 4 QCF) are employed to fill
short gaps in 9 station records from READER. The more uncertain GHCNm QFE (pairwise homogeneity
estimations using adjusted quality-controlled adjacent stations) values are used to fill extended periods with no
observations for Davis, Syowa, and Vostok stations based on adjacent station observations.  All resulting station
temperature records were examined for discontinuities, but none were found.

The details of the spatial extrapolation method using ordinary kriging is paraphrased from Nicolas and Bromwich
(2014). For each month ($t$), the temperature anomaly $\hat{A}(x,t)$ estimated at each point of the grid ($x$) covering
continental Antarctica is derived from a linear weighted combination of the anomalies $A(i,t)$ observed at each of the
15 stations (denoted by $i$), according to the following Eq. (1):

$\hat{A}(x,t) = \sum_{i=1}^{15} \eta_i W_i(x) A(i,t).$                                                                                      (1)

$W_i(x)$ is the weight at point $x$ of the temperature anomaly observed at station $i$ relative to the 1981-2010 mean and
is equal to the square of Pearson's correlation coefficient between the ERA5 2-m air temperature anomaly at the
station and that at grid point $x$ with respect to the 1981-2010 mean after linearly detrending ERA5. The weights are
optimized to minimize the estimation error by accounting for the covariances between the $i$ station records. The
station anomalies $A(i,t)$ are divided by their standard deviation (1981-2010) for normalization and to account for
the spatial differences in variance. $\eta_i$ accounts for the positive (+1) or negative (-1) sign of temperature correlation
between the normalized station anomaly $A(i,t)$ and that at location $x$. The equation yields an estimated normalized
temperature anomaly at each grid point $(x)$ that is then multiplied by the ERA5 temperature standard deviation
(1981-2010) at that point to yield the estimated temperature anomaly.

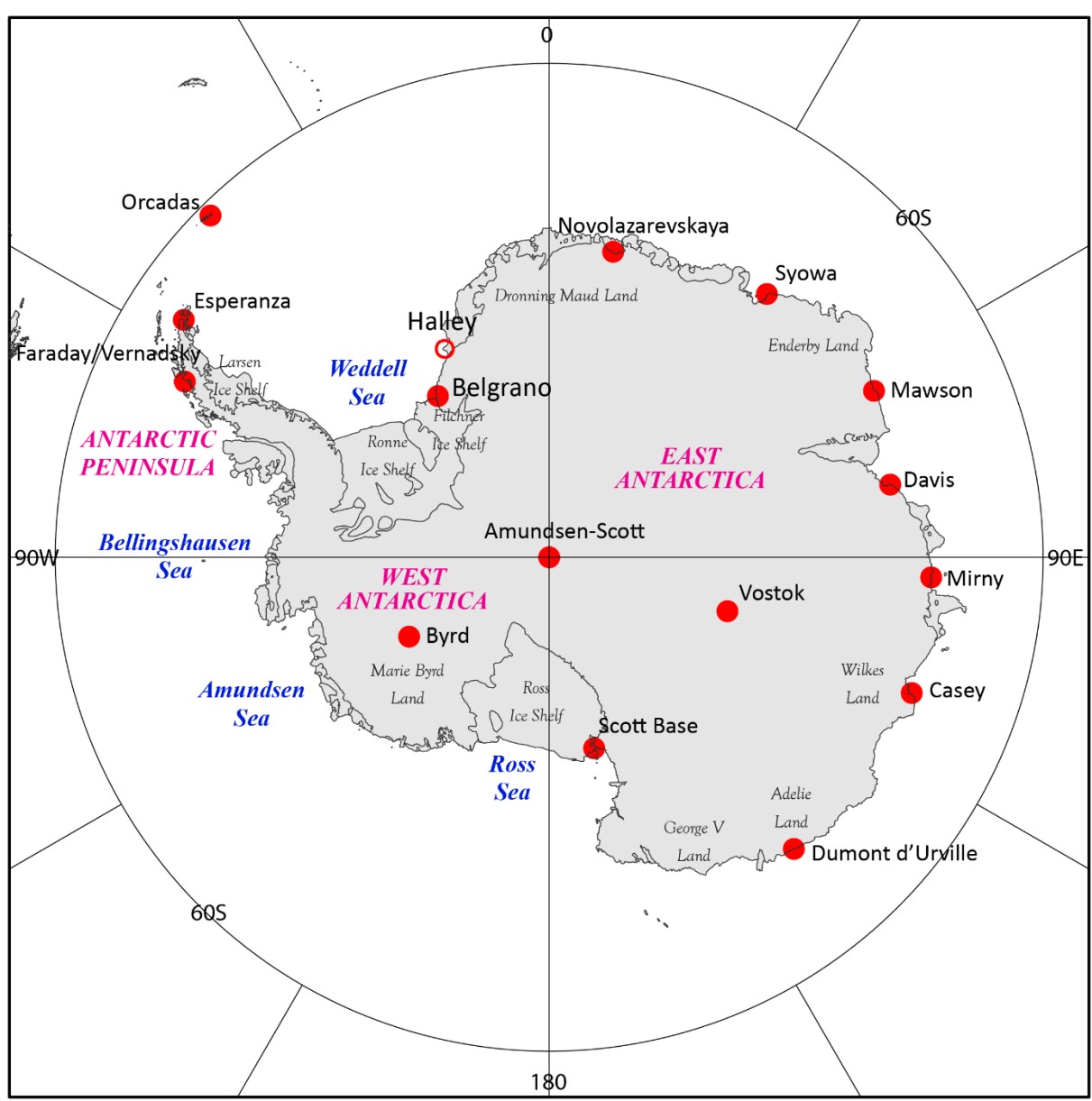

**Figure 1: Antarctic stations (red filled dots) used to reconstruct the near surface air temperature over**
**Antarctica at monthly intervals from 1958-2022. Halley Station (red circle) that is replaced by Belgrano is**
**also shown.**

ERA5 weights and the updated station records were employed to produce the updated Nicolas and Bromwich (2014)
data set that now spans the 66 years from 1958-2022 at monthly intervals: it is called RECON (Bromwich and
Wang, 2024) for the remainder of this manuscript. ERA5 weights were calculated for the 13 available stations in
1958 (Syowa and Novolazarevskaya observations missing), 14 stations in 1959-1960 (Novolazarevskaya
observations missing), and all 15 for 1961-2022 see Table 1.


| Station | Other Observations | GHCNm QFE[2] | Missing |
|---|---|---|---|
| Belgrano II | | 1961-1979, 1980(01-09), 1981(01-05), 2002(03-12), 2003(01-03) | |
| Byrd | 1958-2022[3] | | |
| Casey | 1958(01-12)[1], 1959(01)[1] | | |
| Davis | 2016(05)[1] | 1964(11,12), 1965(01-12), 1966(01-12), 1967(01-12), 1968(01-12), 1969(01,02) | |
| Dumont d'Urville | 2014(04,11)[4], 2015(01-06,09,10)[4], 2016(01,03-06,08-12)[4], 2017(08)[1], 2019(03,04)[1], 2021(05)[1] | | |
| Esperanza | 1979(01,03)[1], 2020(05)[1] | 1979(09,10,11,12) | |
| Faraday/ Vernadsky | 2018(09)[1] | | |
| Mawson | | | |
| Mirny | 2007(02,03,05,06)[1], 2009(06)[1] | | |
| Novolazareyskaya | 2009(10)[1] | 1961(01) | 1958(01-12), 1959(01-12), 1960(01-12) |
| Orcadas | 2002(03-12)[1], 2003(01-12)[1], 2009(02)[1], 2020(05)[1], 2021(04-12)[1], 22(01,02,06,08-11)[1] | | |
| Scott Base | 2016-2022[6] | | 1994(01,02) |
| South Pole | 1958-2022[5] | | |
| Syowa | | 1962(02-12), 1963(01-12), 1964(01-12), 1965(01-12), 1966(01) | 1958(02-12), 1959(01) |
| Vostok | 2007(02,03,05,06)[1], 2009(06)[1] | 1962(01-12), 1963(01), 1994(02-11), 1996(01-12), 2003(02-12), 2004(01,02) | |

| | |
|---|---|
| Primary Source: *UK BAS, Met READER* | |
| Other Data Sources: | [1]**GHCNm v4 QCF** |
| | [2]**GHCNm v4 QFE** |
| | [3]**OSU Polar Meteorology Group** |
| | [4]**Meteo France** |
| | [5]**University of Wisconsin-Madison** |
| | [6]**The National Institute of Water and Atmospheric Research Ltd (NIWA), NZ** |


**Table 1: Data sources employed to fill gaps in the READER data sets to produce RECON. The filled months are indicated by YYYY(MM) or YYYY(MM-MM) format.**

**3. Tests of the Temperature Reconstruction Against Independent Observations**

To validate RECON for Antarctica, we first confirm that RECON fits closely with the observed monthly temperatures at the stations used in the reconstruction, also termed anchor stations. Table 2 demonstrates for those 15 locations (shaded) the correlation is close to 1 at all sites, the bias is minimal, and the root mean square error (RMSE) is small. Thus, RECON reproduces the observations at the anchor sites with high skill.

Next we compare RECON's monthly values against observed monthly temperatures from the READER site not used in the reconstruction (Table 2, unshaded). Almost all of these are from the period after 1979 and are selected so that 10 or more years of continuous observations are available. Stations immediately adjacent to the anchor sites have been excluded and regions with numerous sites, like Terra Nova Bay, are represented by one site. Spatial coverage of Antarctica was also a consideration. 28 of the 57 stations used by Nicolas and Bromwich (2014) were examined along with the addition of 6 new sites. Of the 34 independent stations employed here, only 6 are staffed with the rest being automatic weather stations. On average the bias is tiny and the correlation between RECON and the observations is 0.75. Looking at the individual stations, the correlation is 0.7 or larger at 25 of the 34 independent stations, and the absolute bias is much smaller than 1°C at 32 of the 34 stations. Dome A, the highest point on the East Antarctic Ice Sheet, is a particularly challenging location where surface winds are controlled by the synoptic scale circulation and record low air temperatures can occur in winter (Scambos et al., 2018). Overall, the skill statistics for the current reconstruction using independent observations primarily for 1979-2022 are comparable to those given by Nicolas and Bromwich (2014) for their 54 stations for 1979-2012 with a much smaller RMSE but a smaller correlation coefficient.

An additional test of RECON skill at the observation sites is provided in Table 2 by the $R^2$ metric employed by Nicolas and Bromwich (2014) and defined in their Appendix. This quantity measures the fraction of observational variance explained by RECON and the entire reconstruction period is considered. Table 2 shows that $R^2$ averages 0.98 for the anchor sites and 0.52 for the independent sites. That is, RECON explains on average 52% of the observational anomaly variance at the independent sites. Individual problematic sites where $R^2$ is low are Dome A especially, Relay Station, and Troll. The average $R^2$ for independent stations obtained here for 1979-2022 is significantly smaller than that given by Nicolas and Bromwich (2014, Table 3) for 1979-2012, perhaps a consequence of the large recent atmospheric variability (e.g., Siegert et al., 2023).

| Station | Data Coverage | Lat | Lon | Elevation (m) | Bias | RMSE | r | R$^2$ |
|---|---|---|---|---|---|---|---|---|
| Byrd | 1958-2022 | -80.10 | -119.32 | 1530.0 | 0.00 | 0.27 | 1.00 | 0.99 |
| Faraday | 1958-2022 | -65.10 | -65.15 | 11.0 | -0.03 | 0.31 | 0.99 | 0.98 |
| Orcadas | 1958-2022 | -60.40 | -44.44 | 6.0 | 0.01 | 0.20 | 1.00 | 0.99 |
| Casey | 1958-2022 | -66.30 | 110.50 | 42.0 | 0.02 | 0.37 | 0.99 | 0.97 |
| Scott Base | 1958-2022 | -77.50 | 166.45 | 16.0 | 0.00 | 0.15 | 1.00 | 1.00 |
| Davis | 1958-2022 | -68.60 | 78.00 | 13.0 | 0.00 | 0.34 | 0.99 | 0.97 |
| Mawson | 1958-2022 | -67.60 | 62.90 | 16.0 | -0.01 | 0.31 | 0.99 | 0.97 |
| South Pole | 1958-2022 | -90.00 | 0.00 | 2835.0 | 0.01 | 0.17 | 1.00 | 1.00 |
| Dumont d'Urville | 1958-2022 | -67.00 | 140.00 | 43.0 | 0.01 | 0.22 | 0.99 | 0.99 |
| Mirny | 1958-2022 | -66.50 | 93.00 | 30.0 | 0.02 | 0.53 | 0.97 | 0.93 |
| Syowa | 1959-2022 | -69.00 | 39.35 | 21.0 | 0.00 | 0.20 | 0.99 | 0.99 |
| Esperanza | 1958-2022 | -63.20 | -56.59 | 13.0 | 0.02 | 0.29 | 0.99 | 0.99 |
| Novolazarevskaya | 1961-2022 | -70.50 | 11.49 | 119.0 | 0.00 | 0.04 | 1.00 | 1.00 |
| Vostok | 1958-2022 | -78.50 | 106.90 | 3490.0 | 0.01 | 0.18 | 1.00 | 1.00 |
| Belgrano II | 1958-2022 | -77.90 | -34.60 | 256.0 | -0.01 | 0.54 | 0.97 | 0.91 |
| Elaine | 1993-2022 | -83.10 | 174.20 | 60.0 | 0.07 | 1.58 | 0.75 | 0.50 |
| Lettau | 1986-2022 | -82.50 | -174.40 | 55.0 | 0.03 | 1.67 | 0.78 | 0.55 |
| Bellingshausen | 1968-2022 | -62.20 | -58.90 | 16.0 | -0.02 | 0.95 | 0.85 | 0.58 |
| Fossil Bluff | 2005-2022 | -71.30 | -68.50 | 66.0 | 0.01 | 1.18 | 0.70 | 0.39 |
| Leningradskaya | 1972-1991 | -69.50 | 159.40 | 304.0 | -0.07 | 0.72 | 0.68 | 0.44 |
| Molodezhnaya | 1963-1999 | -67.70 | 45.90 | 40.0 | 0.09 | 0.66 | 0.85 | 0.72 |
| Neumayer | 1981–2022 | -70.70 | -8.40 | 50.0 | -0.05 | 1.51 | 0.62 | 0.35 |
| Rothera | 1976–2022 | -67.50 | -68.10 | 32.0 | -0.04 | 0.98 | 0.89 | 0.75 |
| Russkaya | 1980–1990 | -74.80 | -136.90 | 124.0 | 0.06 | 0.66 | 0.84 | 0.68 |
| Butler Island | 1980–2022 | -72.20 | -60.20 | 91.0 | -0.09 | 1.54 | 0.57 | 0.28 |
| Cape Philips | 1980–2022 | -73.10 | 169.60 | 310.0 | 0.00 | 0.86 | 0.69 | 0.47 |
| D-47 | 2009-2022 | -67.40 | 138.70 | 1560.0 | 0.02 | 0.87 | 0.77 | 0.58 |
| Dome A | 2005-2022 | -80.40 | 77.40 | 4048.0 | -0.98 | 2.75 | 0.44 | 0.01 |
| Dome C II | 1996-2022 | -75.10 | 123.40 | 3280.0 | 0.02 | 1.18 | 0.72 | 0.44 |
| Drescher | 1992–2003 | -72.87 | -19.03 | 34.0 | 0.02 | 0.86 | 0.52 | 0.25 |
| Elizabeth | 1996–2012 | -82.60 | -137.10 | 549.0 | 0.04 | 1.08 | 0.84 | 0.64 |
| GC41 | 1984–2005 | -71.60 | 111.30 | 2763.0 | 0.02 | 1.54 | 0.67 | 0.36 |
| GF08 | 1986–2007 | -68.50 | 102.10 | 2125.0 | -0.02 | 1.21 | 0.75 | 0.49 |
| Gill | 1985-2022 | -80.00 | -178.60 | 30.0 | 0.05 | 1.66 | 0.85 | 0.60 |
| Harry | 1994–2022 | -83.00 | -121.40 | 954.0 | 0.06 | 0.76 | 0.89 | 0.79 |
| Larsen Ice Shelf | 1995–2022 | -66.90 | -60.90 | 17.0 | -0.12 | 1.44 | 0.68 | 0.44 |
| LG10 | 1993-2004 | -71.30 | 59.20 | 2619.0 | 0.02 | 0.54 | 0.80 | 0.62 |
| LG20 | 1991-2004 | -73.80 | 55.70 | 2743.0 | 0.02 | 0.53 | 0.83 | 0.67 |
| LG35 | 1994-2007 | -76.00 | 65.00 | 2345.0 | -0.02 | 0.52 | 0.85 | 0.71 |
| LG59 | 1994-2003 | -73.50 | 76.78 | 2565.0 | 0.00 | 0.44 | 0.83 | 0.68 |
| Law Dome Summit | 1987-1997 2003-2010 | -66.70 | 112.70 | 1368.0 | 0.04 | 0.73 | 0.79 | 0.62 |
| Limbert | 1995–2022 | -75.40 | -59.90 | 40.0 | -0.11 | 1.25 | 0.76 | 0.37 |
| Manuela | 1984–2022 | -74.90 | 163.70 | 80.0 | 0.13 | 0.86 | 0.75 | 0.55 |
| Marble Point | 1980–2022 | -77.40 | 163.70 | 120.0 | 0.06 | 0.62 | 0.94 | 0.89 |
| Marilyn | 1987–2022 | -80.00 | 165.10 | 75.0 | -0.06 | 0.88 | 0.89 | 0.77 |
| Mount Siple | 1992-2005 | -73.20 | -127.10 | 30.0 | -0.06 | 0.69 | 0.74 | 0.54 |
| Relay Station | 1995–2022 | -74.00 | 43.10 | 3353.0 | 0.57 | 1.64 | 0.71 | 0.13 |
| Theresa | 1994–2022 | -84.60 | -115.80 | 1463.0 | 0.09 | 0.83 | 0.77 | 0.57 |
| Troll | 2010-2019 | -72.00 | 2.50 | 1284.0 | 0.16 | 0.86 | 0.58 | 0.17 |
| **Avg. (15 stations)** | | | | | **0.002** | **0.275** | **0.99** | **0.98** |
| **Avg. (34 ind. stations)** | | | | | **-0.001** | **1.060** | **0.75** | **0.52** |

170

*Shading indicates stations that are used to develop the temperature reconstruction RECON. Several independent stations have missing data longer than 12 months.*

**Table 2: Bias, Root-mean-square deviation (RMSE), Pearson's correlation coefficient (r), and fractional observational variance explained by the reconstruction ($R^2$) between temperature reconstruction dataset and station observations with longer-term records, including 34 independent stations. Monthly anomalies are employed.**

To confirm that RECON reproduces the observed long-term 2-m air temperature trends at independent stations that are not used to generate it, a selection of 10 sites with records mostly exceeding 35 years has been made (Table 3). All are coastal or near sea level locations apart from the interior Dome C II record that spans 27 years. All but three locations started after 1979. Modest infilling of the AWS monthly data has been done to produce complete time series. The 4 staffed stations at the top of the table have nearly complete records. Extended periods of comparison are used so that any trends are less likely to be totally swamped by the variability. Table 3 presents the annual temperature trends at the selected stations as well as those from RECON and ERA5. ERA5 is included because it is used next in an example application. It is seen that RECON reasonably captures the trends at all selected sites, and on average does better than ERA5, although variability challenges all these comparisons. The comparison for Neumayer confirms the erroneous ERA5 warming in that region (Bromwich et al., 2024). We therefore conclude that RECON on average captures long-term temperature trends across Antarctica, implying the RECON is appropriate for large-scale analyses. The large spacing between the anchor stations also indicates that localized features would not be resolved. The spatial averaging of RECON is also consistent with the decreased temporal variability that leads to an improved focus on temporal trends.

| Station | Data Coverage | OBS | | RECON | | ERA5 | |
|---|---|---|---|---|---|---|---|
| | | Trend | CI (95%) | Trend | CI (95%) | Trend | CI (95%) |
| Molodezhnaya | 1963-1999 (37) | -0.02 | 0.18 | 0.02 | 0.19 | 0.11 | 0.21 |
| Rothera | 1977-2022 (46) | 0.49 | 0.26 | 0.34 | 0.16 | 0.42 | 0.20 |
| Neumayer | 1981-2022 (42) | -0.11 | 0.16 | 0.01 | 0.09 | 0.54 | 0.18 |
| Bellingshausen | 1968-2022 (55) | 0.20 | 0.12 | 0.29 | 0.14 | 0.26 | 0.12 |
| Lettau | 1986-2022 (37) | 0.10 | 0.35 | 0.15 | 0.17 | 0.49 | 0.37 |
| Marble Point | 1980-2022 (43) | 0.30 | 0.21 | 0.17 | 0.19 | -0.04 | 0.23 |
| Manuela | 1985-2022 (38) | 0.35 | 0.18 | 0.15 | 0.13 | 0.62 | 0.19 |
| Gill | 1985-2022 (38) | 0.23 | 0.33 | 0.10 | 0.16 | 0.44 | 0.34 |
| Dome C II | 1996-2022 (27) | 0.20 | 0.40 | 0.12 | 0.20 | 0.17 | 0.42 |
| Butler Island | 1990-2022 (33) | -0.12 | 0.35 | 0.18 | 0.11 | -0.16 | 0.37 |
| **Average** | | 0.16 | 0.14* | 0.15 | 0.07* | 0.29 | 0.19* |

**Table 3: 2-m air temperature trend comparison (°C/decade) between observations, RECON, and ERA5 at stations with records mostly exceeding 35 years. Number of years entered in parentheses next to record duration. Asterisk values are 95% confidence intervals for average trends. Locations listed in Table 2.**

199

## 4. Example Application

201

Turner et al. (2016) reported that the Antarctic Peninsula region started to cool in 1998 especially during austral summer after decades of warming. Tropically forced decadal variability was inferred to be the cause. The warming apparently resumed in the late 2010s (Carrasco et al., 2021). Similarly, the long-term warming over West Antarctica (Bromwich et al., 2013, 2014) was interrupted around the same time (Zhang et al., 2023). Xin et al. (2023) extracted the primary modes of Antarctic temperature change from 6 reanalyses and 26 observations and noted a marked change in the temperature regime took place around 2000. Figure 2 presents the annual and seasonal continental temperature trends for 1998-2022 according to RECON and ERA5. The annual depictions are broadly similar with some notable differences. The northern Antarctic Peninsula is warming strongly in ERA5 while RECON has trends near zero. ERA5 has marked cooling over the Filchner-Ronne Ice Shelf while RECON finds modest warming, both of which are statistically significant in some regions. As a result of Byrd Station observations and less RECON variability, the annual cooling over West Antarctica is much more marked (and statistically significant) in RECON than ERA5. ERA5's annual warming in Enderby Land is double that of RECON with both being statistically significant. The seasonal trends have a similar pattern but the ERA5 amplitudes are much larger. To ensure the reliability of these results, seasonal and annual trends at the 15 anchor stations were computed for ERA5, RECON, and the observations (not shown). In general, RECON trends were much closer to the observational ones than ERA5 and ERA5 often had significantly larger trends. In addition, ERA5 contains three warming hotspots that continue to 2022 and are artifacts (Bromwich et al., 2024). The results from Table 3 and the findings outlined in this paragraph indicate that the real world more closely follows the RECON depiction than that provided by ERA5.

220

The above results indicate that a comprehensive comparison between RECON and ERA5 Antarctic temperature trends is needed. In addition, contrasting RECON trends with those from the global reconstructions by GIS-TEMP, HadCRUT5, NOAA GlobalTemp, and Berkely Earth would reveal the strengths and weaknesses of each reconstruction. To the extent possible, the causes of the differences found by these comparisons should be identified.

225

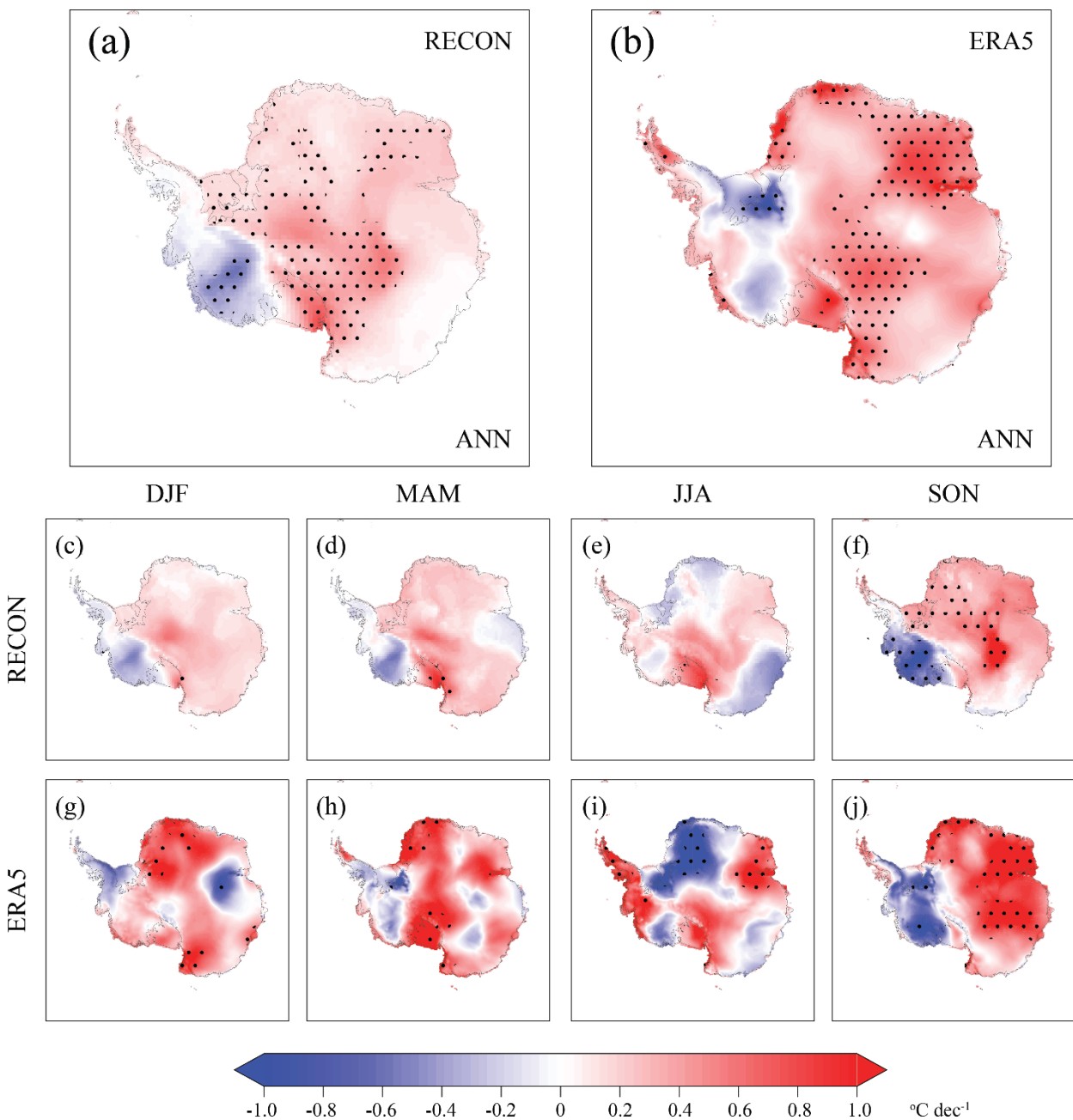

**Figure 2: Annual linear temperature trends from RECON (a) and ERA5 (b) for 1998-2022; seasonal results for RECON (c-f) and ERA5 (g-i) are presented below these. The dots indicate statistical significance of the linear trends at the 0.01 level after considering the lag-1 autocorrelation after Santer et al. (2000).**

**5. Data and Software Availability**

READER**:** https://legacy.bas.ac.uk/met/READER/

Global Historical Climate Network: https://www.ncei.noaa.gov/products/land-based-station/global-historical-climatology-network-monthly

OSU Polar Meteorology Group: Byrd Station: https://polarmet.osu.edu/datasets/Byrd_recon/
Meteo France: Dumont d'Urville: https://meteofrance.com/
University of Wisconsin-Madison: South Pole: https://amrdcdata.ssec.wisc.edu/dataset/amundsen-scott-south-pole-
station-climatology-data-1957-present-ongoing
National Institute for Water and Atmospheric Research: Scott Base: https://cliflo.niwa.co.nz
RECON data described in this manuscript can be accessed at the Antarctic Meteorological Research and Data Center
under https://doi.org/10.48567/efwt-jw56 (Bromwich and Wang, 2024)
The software used to create RECON is available here: https://github.com/shwang-met/Antarctic_Recon
Figures were created using the NCAR Command Language: http://dx.doi.org/10.5065/D6WD3XH5

**6. Discussion**

A reconstruction of Antarctic near-surface air temperatures at monthly intervals for 1958-2022 is presented
(Bromwich and Wang, 2024). It is an update of an earlier data set produced by Nicolas and Bromwich (2014) and
shows skill in reproducing temperature trends at independent stations. The reconstruction is intended for Antarctic
temperature trend analysis for large space and long-time scales and will be compared with other depictions of the
trends in future work. Alternative approaches like statistical downscaling will be needed to produce Antarctic
temperature trends from RECON for regions of complex terrain.

Some desirable improvements can be identified. The southeast Weddell Sea needs a more robust record than
presented here for Belgrano II that has significant infilling. Perhaps the best solution is to homogenize the presently
inhomogeneous Halley temperature record. As shown by King et al. (2021) this will require removing the impact of
the spatial temperature gradients on the Brunt Ice Shelf from the Halley temperature record that comes from varying
station locations and will take significant effort to achieve. Xin et al. (2023) concluded that summer warming over
interior Antarctica may be related to radiative effects of stratospheric ozone and thus be a special environment. Also,
Xie et al. (2023) found from ERA5 for 1958-2020 that Antarctic surface warming amplifies with elevation; this
result is uncertain because of major artifacts in ERA5 especially prior to 1979 (Bromwich et al. 2024). These two
findings suggest that further testing of the RECON trends is desirable for those parts of the East Antarctic plateau
remote from South Pole and Vostok stations to see whether the issues at Dome A and to a lesser extent at Relay
Station (Table 2) are localized.

**Author contributions**
DHB designed the project, wrote the manuscript, and oversaw the analysis. SHW produced RECON data and
performed the analysis with important contributions from XZ and AE.

**Competing interests**
The authors declare that they have no conflict of interest.

**Acknowledgements**

This research was funded by National Science Foundation (NSF) grant 2205398 to D.H.B. X.Z. appreciates the support from NSF grants 2229392 and 2331992.

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
