# Peer review of "An Updated Reconstruction of Antarctic Near-Surface Air Temperatures at Monthly Intervals Since 1958"

_Earth System Science Data, 2024_

## Referee Comment (RC1)

**Review of "An Updated Reconstruction of Antarctic Near-Surface Air Temperatures at Monthly Intervals Since 1958" by David Bromwich et al.**

**Overall Assessment**

The study describes a monthly reconstruction product of Antarctic near-surface air temperature from 1958 to near present-day, which the authors refer to as 'RECON'. It is not entirely novel as it builds upon a similar product developed in an earlier study (Nicolas and Bromwich, 2014), although it appears to be a clear advance on this previous iteration. Broadly, it leverages the few (15) long-term, reliable records that exist (with infilling of data gaps first), in conjunction with spatial extrapolation according to a kriging method, using weights derived from the ERA5 reanalysis. A key difference is that CFSR reanalysis was used instead in Nicolas and Bromwich (2014). These weights are essentially the square of the correlation coefficients between the anomaly of monthly temperatures at each station (after linear detrending) and the monthly anomalies according to ERA5 at each grid point across the continent over the 1981-2020 period. Validation of the algorithm is performed at the locations of the long-term stations, revealing the very high correspondence and close agreement compared to the observations, with moderately strong correlations according to a series of other shorter record stations and automatic weather stations that are completely independent.

The importance of RECON is clear as long-term, reliable records across Antarctica from in situ observations are sparse and reanalyses such as ERA5 are not considered particularly reliable prior to 1979 (which marks the start of the satellite era). Even thereafter, artefacts or discontinuities exist due to several reasons (e.g., temporal irregularities in satellite products assimilated), so the comprehensive coverage of ERA5 cannot always be relied upon for accurate monitoring of temperature trends and variability. The RECON dataset convincingly provides a more accurate picture of long-term monthly air temperature trends, particularly on a more regional scale. However, it is likely that RECON does not necessarily improve our understanding at more local scales, as evidence by closer agreement between observations and ERA5 in some cases, highlighting an important caveat that warrants further attention (perhaps in a follow-up study or using other complementary approaches).

I judge that this paper warrants prompt publication in ESSD and only have a few comments that the authors may wish to consider before acceptance, which I hope they find helpful.

**General Comments**

- The introduction is notably short and does not give any indication of how the rest of the paper is structured. The authors may wish to consider adding this?

- I think the authors could consider discussing their results in the context of what other complimentary approaches would yield, which would make for a more wellrounded paper. Examples could include statistical and dynamical downscaling (using RCMs driven by reanalysis, such as available from CORDEX). Particularly, the limitation of RECON at a localized scale could be addressed via these avenues?

- I do note a major omission that there is no code provided in generating the NetCDF file of the RECON output. This would be good to provide in ensuring the dataset has been generated robustly and consistently with that specified in the paper.

**Specific Comments**

**L13-15:** "It is based on monthly mean 2-m temperatures at 15 fixed stations that are spatially extrapolated to the entire continent using weights derived from the European Centre for Medium-Range Weather Forecasts 5th generation reanalysis (ERA5)". → What is the grid resolution of the derived RECON product (equal to ERA5 at 0.25°?). I found this detail lacking and think such detail should be added here explicitly.

**L23:** "For those regions of Earth that are remote and sparsely populated, establishing their temperature history from direct observations can be a major challenge". → See general comment. Are there other examples from the literature where other similar approaches have been used to help overcome this. It would be good I think for the reader to have a sense of where and on what scale such approaches have been executed before and the relative degree of success, measured in terms of independent, observation-based validation. What were the limitations, and have they been factored into the choices made in the authors' study?

**Table 1:** I find some of this information on data sources used for infilling to be hard to follow. For instance, the GHCN QCF is considered under 'Other Observations' and GHCN QFE is provided as a separate column. Also, how is data made available from the other sources that is not present in MET READER (I thought this data source includes at least some of these such as University of Wisconsin-Madison)?

**L144:** "Overall, the skill statistics for the current reconstruction dominantly for 1979-2022…" → I am not clear what this sentence is conveying. I find the use the word 'dominantly' troubling.

**L152-154:** Is it maybe a little surprising that the average $R^2$ for independent is a little lower versus Nicolas and Bromwich (2014)? Perhaps an additional sentence could suggest why this is (is it just a longer timeframe considered or inclusion of a few more 'problematic sites').

**Figure 2 Caption:** The ERA5 and RECON panels for the annual trends are reversed with respect to the figure caption. The ordering of the seasonal trends is not mentioned for ERA5 and RECON, so this also needs adding.

**Technical Corrections**

**L59:** Double spacing after '(ERA5; Hersbach et al., 2020)'.

**L139:** Full stop missing after 'automatic weather stations'.

**L193:** Double spacing after '2000.'

**L235:** 'longtime scales' → 'long timescales'.

---

## Author Comment (AC1)

**Thanks to both reviewers for valuable input that significantly improved the manuscript.**

**Reviewer 1:**

*General Comments*

• The introduction is notably short and does not give any indication of how the rest of the paper is structured. The authors may wish to consider adding this? **Added.**

• I think the authors could consider discussing their results in the context of what other complimentary approaches would yield, which would make for a more well-rounded paper. Examples could include statistical and dynamical downscaling (using RCMs driven by reanalysis, such as available from CORDEX). Particularly, the limitation of RECON at a localized scale could be addressed via these avenues? **A paragraph has been added to the introduction to highlight the contrasts between RECON and global analyses and a comment about the need for statistical downscaling for regions of complex terrain has been added to the last section.**

• I do note a major omission that there is no code provided in generating the NetCDF file of the RECON output. This would be good to provide in ensuring the dataset has been generated robustly and consistently with that specified in the paper. **Now provided at https://github.com/shwang-met/Antarctic_Recon**

*Specific Comments*

L13-15: "It is based on monthly mean 2-m temperatures at 15 fixed stations that are spatially extrapolated to the entire continent using weights derived from the European Centre for Medium-Range Weather Forecasts 5th generation reanalysis (ERA5)". → What is the grid resolution of the derived RECON product (equal to ERA5 at 0.25°?). I found this detail lacking and think such detail should be added here explicitly. **Added.**

L23: "For those regions of Earth that are remote and sparsely populated, establishing their temperature history from direct observations can be a major challenge". → See general comment. Are there other examples from the literature where other similar approaches have been used to help overcome this. It would be good I think for the reader to have a sense of where and on what scale such approaches have been executed before and the relative degree of success, measured in terms of independent, observation-based validation.

What were the limitations, and have they been factored into the choices made in the authors' study? **See General Comments number 2.**

Table 1: I find some of this information on data sources used for infilling to be hard to follow. For instance, the GHCN QCF is considered under 'Other Observations' and GHCN QFE is provided as a separate column. Also, how is data made available from the other sources that is not present in MET READER (I thought this data source includes at least some of these such as University of Wisconsin-Madison)? **Some more reliable data sets than provided by READER are used. We have clarified the contrast between GHCN QCF and GHCN QFE in the text.**

L144: "Overall, the skill statistics for the current reconstruction dominantly for 1979-2022…" → I am not clear what this sentence is conveying. I find the use the word 'dominantly' troubling. **Reworded.**

L152-154: Is it maybe a little surprising that the average R2 for independent is a little lower versus Nicolas and Bromwich (2014)? Perhaps an additional sentence could suggest why this is (is it just a longer timeframe considered or inclusion of a few more 'problematic sites'). **Material added indicating that the longer period may be responsible because of greater recent atmospheric variability.**

Figure 2 Caption: The ERA5 and RECON panels for the annual trends are reversed with respect to the figure caption. The ordering of the seasonal trends is not mentioned for ERA5 and RECON, so this also needs adding. **Corrected.**

Technical Corrections

L59: Double spacing after '(ERA5; Hersbach et al., 2020)'. **Fixed.**

L139: Full stop missing after 'automatic weather stations'. **Fixed.**

L193: Double spacing after '2000.' **Fixed.**

L235: 'longtime scales' → 'long timescales'. **Changed to "long-time" because noun is "scales".**

**Reviewer 2:**

*Major comments*

- The paper does not discuss the source data in any detail, except for the newly-added Belgrano station. A reader may assume that the remaining 14 stations are taken directly from station data but this is not the case (for example, the 2014 paper discusses the Byrd reconstruction at some

length). The paper should be clear about the input station data sets and discuss any potential homogeneity issues with them. (To give one example with which I am familiar, the 'Casey' record is presumably a composite of Casey with pre-1969 Wilkes and it is unclear whether any potential inhomogeneities with that site move have been considered). **More details about the station records have been added. All time series have been checked for evidence of discontinuities, but none have been identified.**

- From the results in Table 3 and Figure 2, it appears that recent warming in ERA5 is substantially greater than in the reconstruction. This is an interesting result and I think is worth more discussion that it gets. It may also be of interest to compare warming rates in the reconstruction with the Antarctic component of major global temperature data sets (e.g. HadCRUT5, NOAAGlobalTemp, GISTEMP, Berkeley Earth). **ESSD is a data publication. This topic will be explored elsewhere in a research publication.**

Other comments

- L29 – 'collected initially for weather forecasting purposes' – this is common for historical climate records everywhere, is there anything specific to the Antarctic which requires elaboration here?

  **Generally, weather record keeping in Antarctica have not been a high priority with many examples of less care taken than in more temperate climates. The reference by Lazzara et al. (2012) for South Pole has been provided as one example.**

- L60-61 – although Gossart et al (2019) found that ERA5 was in general the best-performing reanalysis for temperature over Antarctica, they did find that it did have a warm bias in the cold season over the interior (although less than CFSR). This should be mentioned somewhere; does it have any implications for the results presented here? **Added. This ERA5 winter shortcoming does not seem to impact the trends it depicts.**

- L73-75 – this implies that the pre-1980 'Belgrano' data are in fact a reconstruction from elsewhere – is this correct? This could be made clearer, and it would be useful to get an indication of how far away the data being used in the reconstruction are. **Added.**

- Figure 1 – I think it would be useful to show the location of the Halley site (perhaps in a different colour) so readers can be aware of how Belgrano replaces it. **Provided.**

- L129 - 'The significantly smaller correlation for Orcadas' – presumably the fact that it's an island (and the surrounding oceans are free of sea ice for a significant part of the year) is also relevant here? It's also surprising to me that the R2 metric in Table 2 is very high for Orcadas when it performs less well on the other metrics, is this worth comment? **Error found in the calculations. Now Orcadas performs like the other anchor sites.**

- Figure 2 – the caption says ERA5 is on the left and RECON on the right but the label on the figures themselves is the other way round. **Fixed.**

- L237-247 – this paragraph is more of a discussion than a conclusion, perhaps the section header could be changed? **Changed.**

**ESSD Chief Editor (Ice)**

There are some usability issues with your NetCDF files.

**Regarding data format**

**The Reconstruction data is produced with standard WRF (NCAR Weather Research and Forecasting Model) model grid format with CF compliant compatibility.**

**An online NetCDF CF checker report is included below.**

**All the model grids latitudes/longtitudes and projection information can be found in either variables or Global Attributes.**

**The dataset has been tested on NCL, Python, GrADS, Panopoly, CDO, NCO and Ncview. We fully expect (untested) it will work on MATLAB and R.**

**We are aware that most GIS softwares have some issues with WRF model grid output, because WRF output is not written for conventional GIS utilities.**

**We believe users can easily apply one of the softwares listed above and adapt to their applications.**

**============================**

**NetCDF CF Checker report**

**https://cfchecker.ncas.ac.uk/**

**Uploaded File: recon_t2m_1958-1970_ano.final.nc**

**CHECKING NetCDF FILE: /tmp/tmp6knnsvko.nc**

**====================**

**Using CF Checker Version 4.1.0**

**Checking against CF Version CF-1.6**

Using Standard Name Table Version 87 (2024-11-12T12:27:52Z)

Using Area Type Table Version 11 (06 July 2023)

Using Standardized Region Name Table Version 5 (12 November 2024)

WARN: (2.3): Global attribute WEST-EAST_GRID_DIMENSION: Attribute names should begin with a letter and be composed of letters, digits and underscores

WARN: (2.3): Global attribute SOUTH-NORTH_GRID_DIMENSION: Attribute names should begin with a letter and be composed of letters, digits and underscores

WARN: (2.3): Global attribute BOTTOM-TOP_GRID_DIMENSION: Attribute names should begin with a letter and be composed of letters, digits and underscores

WARN: (2.3): Global attribute WEST-EAST_PATCH_START_UNSTAG: Attribute names should begin with a letter and be composed of letters, digits and underscores

WARN: (2.3): Global attribute WEST-EAST_PATCH_END_UNSTAG: Attribute names should begin with a letter and be composed of letters, digits and underscores

WARN: (2.3): Global attribute WEST-EAST_PATCH_START_STAG: Attribute names should begin with a letter and be composed of letters, digits and underscores

WARN: (2.3): Global attribute WEST-EAST_PATCH_END_STAG: Attribute names should begin with a letter and be composed of letters, digits and underscores

WARN: (2.3): Global attribute SOUTH-NORTH_PATCH_START_UNSTAG: Attribute names should begin with a letter and be composed of letters, digits and underscores

WARN: (2.3): Global attribute SOUTH-NORTH_PATCH_END_UNSTAG: Attribute names should begin with a letter and be composed of letters, digits and underscores

WARN: (2.3): Global attribute SOUTH-NORTH_PATCH_START_STAG: Attribute names should begin with a letter and be composed of letters, digits and underscores

WARN: (2.3): Global attribute SOUTH-NORTH_PATCH_END_STAG: Attribute names should begin with a letter and be composed of letters, digits and underscores
* * *
Checking variable: time
* * ** * *
Checking variable: RECON
* * ** * *
Checking variable: XLAT
* * *
**WARN: (3): No standard_name or long_name attribute specified**

**ERROR: (3.1): Invalid units: degrees latitude**
* * *
**Checking variable: XLON**
* * *
**WARN: (3): No standard_name or long_name attribute specified**

**ERROR: (3.1): Invalid units: degrees longitude**
* * *
**Checking variable: HGT**
* * *
**WARN: (3): No standard_name or long_name attribute specified**

**ERROR: (3.1): Invalid units: meters MSL**
* * *
**Checking variable: LANDMASK**
* * *
**ERROR: Invalid attribute name: _FillValue_original**

**ERROR: (3.1): Invalid units: (0 - 1)**

**ERRORS detected: 5**

**WARNINGS given: 14**

**INFORMATION messages: 0**

---

## Author Response (AR2)

**Response to Comments by Topic Editor, Marc Mallet.**

Three sentences added at the end of Section 4 on desirable future comparisons between RECON and other reconstructions of Antarctic temperature trends. Brief mention of this is also included in Section 6. Some additional minor edits have been added as a result of rereading the manuscript.

Calculation error in Table 2 was confined to Orcadas. Some other values in Table 2 differ slightly from the original submission because everything was recalculated by another co-author.

NetCDF file in the AMRDC Data Repository dated February 28, 2025 is the final version.

3/27/2025

---

## Author Response (AR3)

**Author's Response, 1 April 2025**

Changes made to 27 March 2025 version: Now reference made to the data set in the abstract and text (at 2 locations) as instructed.